# Economics of Rebreeding Nonpregnant Dairy Cows Diagnosed by Transrectal Ultrasonography on Day 25 after Artificial Insemination

**DOI:** 10.3390/ani12060761

**Published:** 2022-03-17

**Authors:** Silviu-Ionuț Borş, Alina Borș

**Affiliations:** 1Research and Development Station for Cattle Breeding, 707252 Iasi, Romania; 2Department of Public Health, Faculty of Veterinary Medicine, University of Life Sciences, 700489 Iasi, Romania; alinasabie@yahoo.com

**Keywords:** dairy cows, early pregnancy diagnosis, profitability, transrectal ultrasonography

## Abstract

**Simple Summary:**

An early and accurate pregnancy diagnosis can efficiently be used to improve the reproductive performances in dairy cows by using synchronization timed artificial insemination (TAI) programs. This is the key to shortening the calving interval, which improves profitability in dairy farms. Thus, this study presents the feasibility of two TAI programs coupled with early pregnancy diagnosis in dairy cows, 25 days after artificial insemination (AI). Many studies have reported the pregnancy rate when using various reproductive programs, but few have looked at the financial benefits of doing so. By using this strategy, we can generate a profitability of 89.6 USD/cow/year. The contribution to net value presents a breakdown of the income over feed cost, replacement cost, reproductive-program cost, and calf value. The benefit in favor of the TAI programs for the cows failing to conceive in this study is most likely due to the additional cost of the income over feed and given hormones.

**Abstract:**

Pregnancy rates of Holstein cows showed a substantial decline in the past years, which caused intensive TAI programs for nonpregnant cows to shorten the period between unsuccessful insemination and the next attempt on the same cow. Although many studies examined the improvement in pregnancy rates following TAI, only a few examined the economic impact of such programs. In this study, we look at the feasibility of reproductive programs that included early pregnancy diagnosis performed by transrectal ultrasonography 25 days after artificial insemination (AI) and TAI of nonpregnant cows. This resulted in the following two TAI programs: a modified OvSynch program with a second PGF2α treatment at 24 h interval (GPPG, *n* = 100) and a modified OvSynch program with an intravaginal progesterone-release device inserted between days 0–7 (PRID + GPPG, *n* = 100). Cows included in the TAI programs recorded an improvement in the cumulative pregnancy rate (67% vs. 53%; 69% vs. 53%) compared to those in which this strategy was not applied (*p* < 0.05). An economic analysis was performed using a decision-support tool to estimate the net present value (NPV; USD/cow/year). The analysis revealed a difference in NPV of 89.6 USD/cow/year between the programs (rebreeding the nonpregnant cows following the TAI program vs. AI at detected estrus). In summary, rebreeding the nonpregnant cows after early negative pregnancy diagnosis (25 days after AI) using this strategy can improve the cumulative pregnancy rate and profitability of dairy farms.

## 1. Introduction

For better reproductive control in dairy cows, an accurate early pregnancy diagnosis is essential. Early pregnancy detection helps the veterinarian identify open animals and rebreed them as soon as possible, which contributes to an efficient breeding program. After the voluntary waiting period, the number of days a cow remains nonpregnant has been linked to lower profitability [1]. Early detection of cows that are unable to conceive can thus be used to increase the profitability of dairy-herd reproductive programs [2,3].

Different methods for determining pregnant status in dairy cattle might be used. Return to estrus [4], rectal palpation of the reproductive tract [5,6], reproductive-tract ultrasound examination [2], milk-progesterone testing [7], and tests for pregnancy-associated glycoproteins (PAGs) in blood or milk [8,9] are examples of such approaches. Moreover, Doppler ultrasonography has started being used in research to estimate the functionality of the corpus luteum for early pregnancy diagnosis [10,11,12,13,14]. Currently, most veterinarians use transrectal ultrasonography to diagnose early pregnancy in cows. An ideal early pregnancy test would have high accuracy, high sensitivity and specificity, and be inexpensive to perform, consisting of a simple cow-side test (usable in field conditions), and establishing pregnancy status quickly [15].

Depending on the veterinarian′s competence, transrectal ultrasonography is a minimally invasive, accurate, and efficient approach for early pregnancy diagnosis that can be performed as early as day 25 after AI. Transrectal ultrasonography can also provide further information on ovarian structures, identify twins, and establish the viability of the fetus [2].

Pregnancy rates in Holstein cows have steadily declined over time, with a current value of around 35% [16,17,18]. To regulate reproduction in lactating dairy cows within commercial dairies, resynchronization systems to prepare nonpregnant cows for subsequent AI must be created and further assessed under these situations [2]. Some studies have reported the pregnancy rate and time to pregnancy when using various reproductive programs [19,20,21], but few have looked at the financial benefits of doing so [21,22]. The net present value of reinseminating nonpregnant Holstein cows after an early negative pregnancy diagnosis, 25 days after AI, has been reported in a few cases. Thus, the goal of this study was to determine the accuracy of ultrasonography in early pregnancy diagnosis (day 25 after AI) and to calculate the efficacy of two TAI programs for nonpregnant cows in terms of cumulative pregnancy-rate improvement and net present value (USD/cow/year).

## 2. Materials and Methods

### 2.1. Animals and Study Design

The research was performed on 300 Holstein cows artificially inseminated after 60 days in milk (DIM) and divided into three groups: one control (C group, *n* = 100) and two experimental (GPPG group, *n* = 100; PRID + GPPG group, *n* = 100). The cows were housed in free-stall barns with concrete floors covered with mattresses and fed a Total Mixed Ration (TMR) twice per day with ad libitum water access, according to the level of milk production and cow size. To keep the animals healthy, standard management practices, including a cooling system coupled with a weather station for hot months, were followed. During the study period, the farm milked about 780 cows three times a day at 0400, 1200, and 1900, for a daily average of 66 lb milk/cow/day. During the study, cows were balanced between pens by DIM and parity. Calving dates, breeding dates, and DIM were obtained from AfiMilk management software (AfiMilk, Kibbutz Afikim, Israel).

Estrus cows were identified from the AfiMilk (AfiMilk, Kibbutz Afikim, Israel) estrus daily report and each one was examined by an experienced veterinarian. The attempt to mount other cows, chasing herd mates, restlessness, chin resting, sniffing herd mates′ vagina and bellowing, congestion, relaxation, and mucus discharge from the vulva were the estrus signs. The manifestation of standing estrus was considered to be a sign of true estrus.

The nonpregnant cows from the GPPG group were subjected to a synchronization TAI protocol to induce ovulation for timed AI (TAI). The TAI protocol started at 25 days after the first AI, which was day 0 of the protocol, with 100 µg Gonadorelin, (Gonavet Veyx, Veyx-Pharma GmbH, Schwarzenborn, Germany). On days 7 and 8 after GnRH, two doses of 500 µg Cloprostenol (PGF Veyx forte, Veyx-Pharma GmbH, Schwarzenborn, Germany) were administered, followed by 100 µg Gonadorelin, (Gonavet Veyx, Veyx-Pharma GmbH, Schwarzenborn, Germany) on day 9 and TAI on day 10 (approximately 16 hr later). The nonpregnant cows from the PRID + GPPG group were subjected to a similar synchronization TAI protocol, with the difference being that intravaginal progesterone-release device PRID delta (Ceva Santé Animale, Loudeac, France) was used in the day interval 0–7 of the protocol. The nonpregnant cows from the C group were not treated and were AI-inseminated at the next spontaneous estrus.

One experienced veterinarian conducted all ultrasound examinations and hormone injections. The transrectal ultrasonography (iScan 2, Draminski S.A., Olsztyn, Poland) was performed 25 days after the first AI and repeated on day 32 for pregnancy confirmation (the reference test). Ultrasound scanning of the uterus and the ovaries was processed using a 3–7.5 MHz rectal convex probe (Draminski S.A., Olsztyn, Poland) for diagnosis and confirmation of pregnancy. The visualization of anechogenic fluid-filled uterine horn (embryonic vesicle more than 10 mm in diameter) in association with a corpus luteum on the ipsilateral uterine horn was used as a positive indicator of the pregnancy.

### 2.2. Economic Analysis

A 780-cow commercial dairy herd with a production of 26,000 lb milk/cow/year was simulated using the UW-DairyRepro$ decision support tool [22] with the modifications described by Giordano et al. [23] to assess the economic impact of rebreeding nonpregnant cows by using the mentioned TAI programs. The reproductive program simulated for the first AI service was similar to the experiment (heat breeding), whereas the second reproductive-management programs compared in the current study used TAI programs initiated 25 days after the first AI in the nonpregnant cows from experimental groups vs. AI at detected estrus in the C group. The following herd, economic, and reproductive parameters are included: average body weight (1600 lb), involuntary culling (28%/yr), mortality rate (4%), stillbirth (4.9%), milk price (16 USD/cwt), cost feed lactation (0.08 USD/lb DM), the dry period fixed cost (0.06 USD/lb DM), female calf value (USD 200), male calf value (USD 100), the heifer replacement value (USD 1800), salvage value (0.526 USD/lb), the adjusted voluntary waiting period (85 d in C group, 87 and 86 d in the GPPG and PRID + GPPG group), estrus cycle duration (29 days in all three groups), maximum day in milk for breeding (300 days), interbreeding interval for TAI service (35 days), pregnancy rate first service (43% vs. 42, and 41%), pregnancy loss (5%), day in gestation first pregnancy check (25 days), day in gestation second pregnancy check (32 days), day in milk first injection for TAI service (112 d in GPPG group, and 111 d in PRID + GPPG group), estrus detection rate (53%). The cumulative pregnancy rates were set at 53% in the C group, 67% in the GPPG group, and 69% in the PRID + GPPG group. The reproductive program cost used for experimental groups included GnRH at 2.6 USD/dose, PGF at 2.6 USD/dose, PRID delta at 15 USD/unit, labor for hormone injections at 15 USD/h, and AI (including semen unit and labor) at 45 USD/AI. The pregnancy diagnosis cost was set at 100USD /h. The model estimated net present value (NPV; USD/cow/year) differences for the reproductive programs consisting in rebreeding the nonpregnant cows following TAI programs vs. AI at detected estrus.

### 2.3. Statistical Analysis

The accuracy of ultrasound examinations used for early pregnancy diagnosis (25 days after AI) was evaluated according to Broaddus and DeVries, [24]: sensitivity (Sn) = TP/(TP + FN) × 100, specificity (Sp) = TN/(TN + FP) × 100, accuracy (Acc) = (TP + TN)/(TP + TN + FP + FN) × 100, positive predictive value (PPV) = TP/(TP + FP) × 100, and negative predictive value (NPV) = TN/(TN + FN) × 100, where TP = no. of animals correctly diagnosed as pregnant, TN = no. of animals correctly diagnosed as nonpregnant, FP = no. of nonpregnant animals declared wrongly as pregnant, and FN = no. of pregnant animals declared wrongly as nonpregnant.

The pregnancy rate at first AI was defined as the percentage of cows that became pregnant after first AI out of the total number of cows in the corresponding group. The cumulative pregnancy rate was defined as the percentage of cows that became pregnant after two rounds of AI out of the total number of cows in the corresponding group, but not more than 35 days after the first AI.

Data were analyzed by least-squares analysis of variance using GLM procedures of SAS (SAS Institute, Cary, NC, USA). The fixed effects in the model included group, parity (primiparous vs. multiparous), and their interaction. A chi-square model was used in the final analysis after the removal of nonsignificant interactions. The differences between the two groups were considered to be statistically significant when *p* < 0.05.

## 3. Results

Pregnancy diagnosis performed 25 days after AI by transrectal ultrasonography showed high accuracy in all three groups (Table 1).

Because the pregnancy rate after the first insemination was nearly identical in the experimental and control groups (42, 41% vs. 43%), approximately 60 percent or fewer of the cows failed to conceive. The corpus luteum was found in a large proportion of nonpregnant cows in this study in all three groups compared with the ovarian follicles and cysts which were detected in a small proportion (Table 1). When nonpregnant cows from the experimental groups were subjected to a synchronized TAI program, they had a higher cumulative pregnancy rate in GPPG and PRID + GPPG groups (67% and 69%) compared with the C group (53%, *p* < 0.05; Table 1).

The simulating program used in this experiment, which included the improvement of the cumulative pregnancy rate by rebreeding the nonpregnant cows as a result of the TAI program, showed an NPV of 89.6 USD/cow/year greater for the experimental groups compared with the C group (Table 2).

## 4. Discussion

This study presents the NPV of two TAI programs for nonpregnant cows, which shortens the period between an unsuccessful AI and the following attempt.

The pregnancy detection by transrectal ultrasonography performed 25 days after AI yielded results comparable to those obtained by others [25,26]. This suggests that on day 25 after AI, transrectal ultrasonography can accurately identify nonpregnant cows eligible for TAI programs. Some studies [27,28] found low sensitivity on day 25 after AI, possibly due to difficulties in visualizing the gestational vesicle, ambiguities in interpreting the ultrasonography image [29], or the embryonic loss between days 25 and 45 after AI. Some errors can occur, and they can be costly: a false-positive diagnosis can result in the open cows not being inseminated on time, whereas a false-negative diagnosis can result in abortion when prostaglandin is administered. In our study, a small number of cows from both groups had false-negative pregnancy diagnoses due to very small embryonic vesicle dimensions (<10 mm) in the uterus. These cows were recorded as nonpregnant, but were not treated since the pregnancy diagnosis could not be certain. On day 32 after AI, the pregnancy was confirmed. Since the cost of an abortion is higher, rebreeding the cows in such ambiguous cases is not justified. When this reproductive management is applied, we believe that the accuracy of transrectal ultrasonography used for pregnancy diagnosis should be around 90–100%. Other authors suggested giving prostaglandin to cows found open on day 35 after AI for rectal palpation and 28 days for the ultrasound exam [30].

Early pregnancy detection is critical for reducing the calving interval by allowing the veterinarian to identify open cows and rebreed them. The pregnancy rate at first AI did not differ between groups in our study, and the findings are consistent with previous research [19,31,32,33,34]. The majority of the authors agree that these low pregnancy rates are most likely the result of a combination of factors, including changes in physiology, nutritional management of the transition period, and the selection of traits that may have adverse effects on fertility [16,35,36,37]. Furthermore, pregnancy rate failures in lactating dairy cattle can be attributed to the lack of a viable embryo due to oocyte quality and fertilization issues or the inability of the uterus to support blastocyst growth and conceptus elongation [24]. As a result of lower pregnancy rates, the dairy industry has made strides in developing estrus synchronization programs for both before and after pregnancy diagnosis.

An interesting aspect of our study is that corpus luteum was found in a high proportion of nonpregnant cows from both groups. Scully et al. [38] found no differences in corpus luteum echotexture measurements between pregnant and nonpregnant cows from day 18 to day 21. With a 90% fertilization rate, embryonic mortality between pregnancy and day 24 is approximately 40%; 70 to 80% of this loss occurs between day 8 and day 16 [39]. Ricci et al. [40] concluded that at least half of the nonpregnant cows who kept their corpus luteum until day 32 after AI were initially pregnant but lost the pregnancy early. However, our study did not evaluate the embryonic losses, because there are several risk factors associated with this, including environmental stress, disease, nutrition, luteal insufficiency, and ovulation of persistent follicles [41,42,43]. As a result, establishing a link between early pregnancy diagnosis and embryonic loss is extremely difficult. Unfortunately, there is little evidence of when an embryonic loss occurs, making it difficult to link it to the mechanism that causes it [44].

The most common measure when cows are found open at pregnancy diagnosis is to apply the treatment with prostaglandin, which results in luteolysis followed by a new estrus, typically about 3–4 days later [45]. Some authors recommended initiating the TAI program even before pregnancy diagnosis [46]. In our study, the TAI programs were initiated on the first day when the nonpregnancy was diagnosed. This reproductive management program, which included starting TAI programs for nonpregnant cows on the day of pregnancy diagnosis (day 25 after AI), resulted in a pregnancy rate of 43.1% (25/58) for the nonpregnant cows from the GPPG group and 47.5% (28/59) for the nonpregnant cows from the PRID + GPPG group, which is comparable to or better than the results obtained by other studies. Pereira et al. [47] obtained pregnancy rates of 26.4% and 20.1% when the TAI protocol was initiated 31 ± 3 days after AI for nonpregnant cows subjected to pregnancy diagnosis by ultrasonography and transrectal palpation, respectively. Moreira et al. [48] found similar results when using an estrus-synchronization protocol beginning 20 days after AI, with a reported pregnancy rate of 20% on day 45 of gestation. Fricke et al. [46] observed a pregnancy rate of 23%, 34%, and 38% for cows enrolled in the synchronization protocol at 19, 26, and 33 days after AI, respectively. El-Zarkouny et al. [49] reached a conception rate of 59.3%, while Bisinotto et al. [50] noted 51.3%, after a protocol with a gestagen insert but also with a double injection of PGF2α on day 7 of the OvSynch program. At the same time, there is controversy regarding the use of tools for early pregnancy diagnosis in combination with estrus-resynchronization programs. Injecting pregnant cows with GnRH 19 days after the first AI does not improve the calving rate compared to starting the Resynch program 26 or 33 days after the first AI [46]. Similar results were obtained when the Resynch program started on day 21 after timed AI (TAI) was used, to initiate a TAI protocol before pregnancy diagnosis [51]. Moreira et al. [48] halted the aggressive estrus-resynchronization program due to the high embryonic loss from day 20 to 27 in pregnant cows treated with GnRH on day 20 after the TAI compared to untreated pregnant cows. In our opinion, the TAI programs should begin on the day of early pregnancy diagnosis (day 25 after AI service) because this strategy can improve the profitability of dairy farms without risks.

The contribution to net value presents a breakdown of the income over feed cost, replacement cost, reproductive-program cost, and calf value. The benefit in favor of the TAI programs for the cows failing to conceive in this study is most likely due to the additional cost of the income over feed and given hormones. This benefit, in favor of the GPPG and PRID + GPPG groups, was assessed as a positive net present value for the farm, and its relationship with additional reproductive-management decisions should be considered. In another study, Pereira et al. [47] began resynchronization 31 ± 3 days after AI and obtained an NPV of 3.65 USD/cow when pregnancy diagnosis was performed on the same day by ultrasonography versus 38 ± 3 days after AI by transrectal palpation. However, the majority of studies using management practices aimed at reducing the interbreeding interval in cows have reported economic benefits, particularly for reproductive programs with lower pregnancy rates [21,22,52]. The contribution to pregnancy rate and net present value could be generated by not supplementing the TAI programs with an activity-monitoring system. Fricke et al. [53] assessed the efficacy of TAI with and without an activity-monitoring system at first service. They found that this combination reduces time to first service by 7.5 to 12.4 days while decreasing the conception rate by 8% when compared with TAI alone. When evaluating the net present value of TAI with or without an activity-monitoring system, they discovered that the difference is only 4.00–8.00 USD/cow/year. The use of an activity-monitoring system to inseminate cows based on increased activity reduced the days to first AI by increasing the AI service rate, whereas cows receiving 100% TAI after completing a TAI program had more P/AI [53]. This suggests that depending on individual herd scenarios, multiple reproductive-management programs may be economically feasible [54]. Thus, in the cattle industry, a variety of strategies based on rebreeding nonpregnant cattle can be used to improve net present values. Because herds differ in reproductive performance [55], management decisions should be based on an economic analysis of observed reproductive outcomes specific to that farm [53].

## 5. Conclusions

In this study, we used transrectal ultrasonography for pregnancy detection as an accurate method of identifying the nonpregnant cows eligible for the synchronization TAI programs, 25 days after AI. It is possible to improve the cumulative pregnancy rate and the net present value by rebreeding the nonpregnant cows as soon as possible. Thus, rebreeding nonpregnant cows starting with day 25 after AI can reduce the time in which a cow becomes pregnant and increase the cumulative pregnancy rate (67% vs. 53%; 69% vs. 53%) and the profitability of dairy farms by around 89.6 USD/cow/year.

## Figures and Tables

**Table 1 animals-12-00761-t001:** Results of the early pregnancy diagnosis, cumulative pregnancy rates, and net present value in GPPG and PRID + GPPG groups compared with the C group.

Variables	C Group	GPPG Group	PRID + GPPG Group
No. of cows scanned	100	100	100
No. of cows nonpregnant at first pregnancy diagnosis (25 days after AI)	55	53	54
No. of cows pregnant at first pregnancy diagnosis (25 days after AI)	45	47	46
Pregnant cows diagnosed as nonpregnant (false negative)	2	1	2
Nonpregnant cows diagnosed as pregnant (false positive)	4	6	7
No of cows pregnant at 32 days after AI	43	42	41
No. of cows nonpregnant at 32 days after AI	57	58	59
Sensitivity of ultrasonography at 25 days after AI	95.6%	97.7%	95.3%
Specificity of ultrasonography at 25 days after AI	93.4%	90.6%	89.4%
Accuracy of ultrasonography at 25 days after AI	94.3%	93.5%	91.7%
Positive predictive value of ultrasonography at 25 days after AI	91.5%	87.5%	85.4%
Negative predictive value of ultrasonography at 25 days after AI	96.6%	98.3%	96.7%
Nonpregnant cows, diagnosed with corpus luteum	75.4%	67.2%	74.6%
Nonpregnant cows, diagnosed with ovarian follicles at different development stages (size, 3–20 mm)	12.3%	17.2%	15.3%
Nonpregnant cows, diagnosed with ovarian cysts (size,>20 mm)	12.3%	15.6%	10.1%
Percentage distribution of cows by parity (%Primiparous/Multiparous)	33/67	47/53	45/55
The average interval between calving and the first pregnancy diagnosis (days, mean ± standard deviation)	110.2 ± 6.1	112.3 ± 7.1	111.5 ± 8.1
Nonpregnant cows treated	0	52	52
Cumulative pregnancy rate	53% ^b^	67% ^a^	69% ^a^
Net value (USD/cow/year)	2433	2522	2522

^a,b^ Different superscripts indicate significant differences between groups (a vs. b, a > b, *p* < 0.05), (chi-square analysis).

**Table 2 animals-12-00761-t002:** Contribution to net value.

Items	Control Group	Experimental Groups	Difference
Total net value (USD/cow/year)	2432.5	2522.1	89.6
Income over feed cost (USD/cow/year)	2623.7	2757.8	134.1
Replacement cost (USD/cow/year)	−224.4	−257.7	−33.3
Reproductive cost (USD/cow/year)	−41.3	37.5	3.8
Calf value (USD/cow/year)	74.5	59.5	−15

## Data Availability

Not applicable.

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
