# Peer review of "Economics of Rebreeding Nonpregnant Dairy Cows Diagnosed by Transrectal Ultrasonography on Day 25 after Artificial Insemination"

_animals, 2022, doi:10.3390/ani12060761_

Round 1

Reviewer 1 Report

Overall comments:

The manuscript “Economics of Rebreeding Non-Pregnant Dairy Cows” presents an economic approach on a reproductive study, and in fact, this is the endpoint on a dairy herd. The author’s emphasis is about one of the major issues on dairy industry, the longer days open. It is proposed an economic analysis based on Day 25 pregnancy test. There are few studies considering the economic profits of rebreeding/resynch.

Originality: The manuscript reports the economic analyses based on Day 25 pregnancy test.

Points to justify:

1) Why the authors believe the Day 25 pregnancy diagnosis is appropriate. If we consider the cow is not pregnant, it will return to estrous before Day 25. Also, besides PAGs to check pregnancy, there are ISGs prior to Day 20, and if absent the cows are open. These few days considering ISGs (not mentioned in the manuscript) and return to estrus would increase profitability?

2) The authors performed an economic analysis based on Day 25 pregnancy diagnosis. I would like to see a prediction on Days 20 and 30 using the same parameters as it was obtained.

3) As it was mentioned in the discussion, embryo mortality is difficult to detect, so the earlier the better. There are also few studies approaching early embryo mortality.

Appropriate conclusions: Conclusions are appropriate to their study. Nevertheless, I would recommend the authors to include the economic conclusion from their data, and state that this is on Day 25 pregnancy diagnosis.

Reviewer 2 Report

Concern : review of the paper entitled : « Economics of Rebreeding Non-Pregnant Dairy Cows »

General comment

I ave had some difficulties to understand the economic impact of such resynchronisation program attending the absence of specific data concerning the considered parameters. The authors are speaking about a simulated program using the following parameter :  26,000lb milk/cow/year, 28% involuntary culling rate, 4% mortality rate, and 4.9% stillbirth rate. What about the performances in this farm ?

Specific comments

  • Line 16 : see also Early pregnancy diagnosis on large dairy farms and its role in improving profitability Fodor I et Ozavari L : Animal welfare, ethology and housing systems, 2018, 14,1,22-36) and de Vries et al., Proceedings 2nd Florida Dairy Road Show, 2005,31–41
  • Line 46 Differents publications have proposed to use echo-Doppler for a very early non pregnancy diagnosis (D21-D24) : see : Matsui et al. 2009, Miyamoto et 2006, Herzog et al. 2010, Siqueira et al. 2013, Pugliesi et al. 2014, Utt et al. 2009, Scully et al. Theriogenology 2014). May be the authors could mentioned such application.
  • Line 50 : could you replace farmers by veterinarian
  • Line 101 : The non-pregnant cows from the C group were not treated and were AI inseminated at the next spontaneous estrus : could you tell the interval between the non pregnancy diagnosis and the next AI ? In the Table 1, you mention that 55 cows (group C) are non pregnant 25 days after first pregnancy diagnosis
  • Line 105 : which diameter has been considered to confirm the présence of a corpus luteum . Indeed the % of identified CL is rather very high : 67 to 75 % (Table 1)
  • Line 117 : The pregnancy rates were set at 53% in the C group, 67% in the GPPG group, and 69% in the PRID + GPPG group based on the results of the first part of the study : where is the description of this first part ?
  • Line 118 : attending you have done two pregnancy diagnosis at day 25 and 32 after AI, why have you not evaluate the embryonic mortality rate during this interval ?
  • Line 123 and 124 : the proposed figures are theorical or observed in the concerned farm ? Why have you simulate a scenario and not calculate the economic impact considering the data collected in the farm ?
  • Line 143 : « The cumulative pregnancy rate was defined as the percentage of cows that became pregnant after two rounds of AI » : why have you limited the results to the first two AI and not to the total number if AI done until the decision of culling ?
  • Line 163 Table 1
    • Positive and not Pozitive ;
    • predictive and not predictiv :
    • Lactating/Dry : how can you explain that dry cow are still in the reproductive herd ?
    • Could you mention the average interval between non pregnancy diagnosis and pregnancy after two AI ?
    • Could you mention for each group the average interval between calving and the pregnancy diagnosis done 25 days after the previous insemination
    • What the average AI number realized in each group before the early pregnancy diagnosis ?
    • of cows non-pregnant at first pregnancy diagnosis (25 days after AI) 53 54 but the number of non-pregnant cows treated were 58 59 : can you explain such difference ?
    • Could you indicate the average interval between calving and pregnancy for each group ?
  • Line 182 : it’s an evidence not ?
  • Line 184 : small vesicle ? which minimal size can be considered to confirm a pregnancy ¿ : the presece of an embryo could be a better parameter but at this stage of pregnancy, his presence is rather difficult to identify.
  • Line 222 : non pregnancy diagnosis and not pregnancy
  • Line 256 : observed reproductive outcomes specific to that farm : I agree but it seems such specific analysis don’t have be done in your case.

Reviewer 3 Report

.

Round 2

Reviewer 2 Report

Thanks to the authors to have taken in account the proposed remarks